# Position: LLM Benchmark Datasets should be Contamination-Resistant

Ali Al-Lawati [1]   Jason Lucas [1]   Dongwon Lee [1]   Suhang Wang [1]

## Abstract

Benchmark datasets are critical for reproducible, reliable, and discriminative evaluation of LLMs. However, recent studies reveal that many benchmark datasets are included in pretraining corpora, i.e., *contaminated*, which diminishes their value as reliable measures of model generalization. In this paper, we argue that benchmark datasets should be *contamination-resistant*, i.e., *unlearnable*, but support *inference*. To accomplish this, we first highlight the wide prevalence of benchmark dataset contamination and outline the properties of contamination-resistant datasets. Second, we highlight how the asymmetry between the inference and training pipelines in the Transformer architecture can be leveraged to support contamination-resistance. Third, we outline mathematical advancements to make these datasets interoperable across various LLM architectures. Based on the above, we call on the community to ensure the reliability of LLM benchmarking by: (i) advancing novel contamination-resistant methodologies, (ii) developing supporting methods and platforms, and (iii) adopting contamination-resistant benchmarks into existing evaluation pipelines.

## 1. Introduction

The growing demand for high-quality data in large language models (LLMs) has motivated developers to aggressively search and utilize nearly all available text data in pretraining corpora (Xu et al., 2024). Given the enormous scale of these corpora, it has become nearly inevitable that benchmark samples are ingested and used in pretraining — a phenomenon commonly referred to as *benchmark dataset contamination* (Singh et al., 2024). As a result, the value of benchmarking is fundamentally diminished, as it is no

longer able to provide a reliable measure of model performance and generalization. Instead, it often reflects the model's capacity to effectively recall data encountered during pretraining, thereby inflating accuracy (Ni et al., 2025; Dong et al., 2024). To illustrate this, Zhang et al. (2024) demonstrated that using a clean mirror set of GSM8K (a non-public version) can reduce accuracy by up to 13% on popular model families such as Mistral.

Even when explicit measures are put in place to exclude known benchmarks (Touvron et al., 2023; Brown et al., 2020), the public availability of benchmark datasets makes them an easy target for repurposing or inclusion in derivative datasets. In this case, data may leak into pretraining corpora indirectly (Li et al., 2024). Moreover, the complex nature of modern LLM training, often encompassing techniques such as distillation and teacher forcing, may further contribute to indirect contamination (Sun et al., 2025).

As a result, benchmark contamination is a significant obstacle to fair LLM evaluation and can quickly render benchmarks obsolete (White et al., 2025). This has been corroborated by various leakage detection methods applied to popular LLMs (Deng et al., 2024). Recent studies have detected widespread contamination across popular LLMs, with levels reaching up to 45% on commonly used benchmarks (Li et al., 2024).

To support reliable LLM benchmarking, researchers have proposed several approaches. For example, one direct approach is to keep benchmark datasets private and rely on a trusted third party to perform evaluations (Rajore et al., 2024). However, while this prevents leakage, it risks stifling innovation by increasing barriers to independent verification. Alternatively, constantly updating benchmarks to overcome this problem, i.e., dynamic benchmarking (White et al., 2025; Wu et al., 2025b), addresses static leakage but requires significant effort and complicates longitudinal comparisons, as a shifting target makes it difficult to maintain a consistent baseline for tracking model progress over time. Researchers have also proposed methods to rephrase benchmark datasets, or decontaminate them by identifying and removing leaked instances (Xu et al., 2024). However, both strategies tend to degrade benchmark utility and difficulty. Moreover, decontamination faces a scale challenge: as corpus sizes grow into trillions of tokens, these methods become increasingly inadequate at accurately identifying

---

[1]The Pennsylvania State University, University Park, PA, USA. Correspondence to: Ali Al-Lawati <aha112@psu.edu>.

*Proceedings of the 43rd International Conference on Machine Learning*, Seoul, South Korea. PMLR 306, 2026. Copyright 2026 by the author(s).

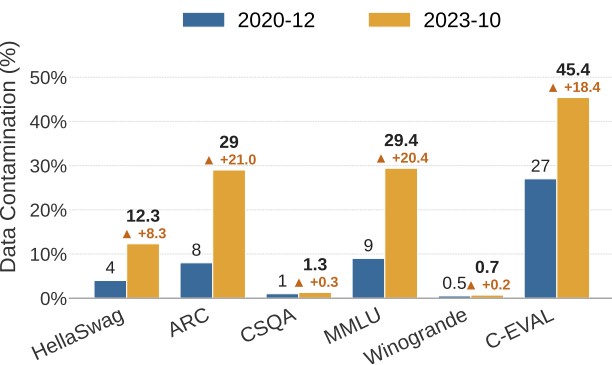

*Figure 1.* Detected contamination from 2020 to 2023 for various benchmark datasets (Figure is from (Li et al., 2024))

leakage instances (Jiang et al., 2024).

> **Position statement**: We advocate for a fundamental shift toward releasing benchmark datasets in a contamination-resistant form, ensuring data remains useful for evaluating models while being unlearnable during public release.

We ground this position in the architectural asymmetry between Transformer (Vaswani et al., 2017) training and inference pipelines, and draw on recent mathematical advances and empirical findings on interoperability across model architectures (Bansal et al., 2021). This ensures that benchmark data remains a reliable measure of model generalization by making it unsuitable for inclusion in pretraining corpora.

This position paper is organized as follows: Section 2 highlights the prevalence of benchmark contamination and defines contamination resistance. Section 3 demonstrates training–inference asymmetry in Transformers. Section 4 addresses interoperability. Section 5 and Section 6 discuss broader implications and alternative views. Finally, we conclude with future directions in Section 7.

## 2. Contamination-Resistance: Motivation and Formalization

### 2.1. Benchmark Contamination Prevalence

Benchmark contamination is a widespread and prevalent issue that undermines reliable evaluation of LLMs (Cao et al., 2025). Li et al. (2024) found that contamination levels have increased over time (see Figure 1). Brown et al. (2020) flagged over 90% of examples in some benchmarks as contaminated during the evaluation of GPT-3. More recently, audits of state-of-the-art systems reveal that even controlled releases like Llama 2 exhibit significant contamination in over 16% of the MMLU suite (Touvron et al., 2023). Ahuja

et al. (2024) reveal that nearly all major LLMs exhibit significant data contamination of up to 91.8% across popular multilingual benchmarks. This effect grows with scale, as larger models have a higher capacity for verbatim memorization. As a result, even a small fraction of contaminated data within a trillion-token training corpus can substantially compromise evaluation integrity and undermine the validity of reported benchmark results (Magar & Schwartz, 2022).

Moreover, due to large-scale web scraping, newly released benchmarks are quickly copied across repositories, discussion forums, and derivative datasets, which makes it difficult to exclude them reliably from new training corpora. As a result, even gated benchmarks (Tanzer et al., 2024) may be indirectly incorporated through data aggregation pipelines, model distillation, or continual pretraining (Sun et al., 2025; Magar & Schwartz, 2022). This shortens the effective lifespan of benchmarks as clean evaluation tools and exacerbates contamination risks, particularly for frontier models trained on frequently refreshed corpora (Samuel et al., 2025).

### 2.2. Making Data Unlearnable

*Unlearnable data* is a technique that prevents machine learning models from extracting useful patterns from data during training (Li et al., 2025). Existing approaches rely on adversarial perturbation to add imperceptible noise that disrupts gradient-based learning (Huang et al., 2021), shortcut learning techniques to embed misleading patterns (Sandoval-Segura et al., 2022), or targeted data poisoning to degrade model performance (Fowl et al., 2021). However, these approaches are largely incompatible with the discrete nature of text. Moreover, modern LLMs are robust denoisers capable of recovering meaning from noisy, transformed, or watermarked text. Simple operations like paraphrasing or back-translation can effectively remove crafted perturbations without changing the underlying semantic content.

### 2.3. Contamination Resistance

To advance our position on contamination-resistant datasets (CRDs) for LLM benchmarking, we first formalize the concept of contamination resistance.

**Definition 2.1** (Contamination-Resistant Dataset (CRD)). A dataset $\phi(\mathcal{D})$ is considered contamination-resistant with respect to a model $\mathcal{M}$ and a transformation function $\phi$ of plaintext data $\mathcal{D}$ if it maintains **inference utility:** $\mathcal{M}(\phi(\mathcal{D}))$ yields valid task performance, while being **unlearnable:** for a loss function $\mathcal{L}$ and model parameters $\theta$, any training steps on $\phi(\mathcal{D})$, $\nabla_\theta \mathcal{L}(\mathcal{M}_\theta, \phi(\mathcal{D}))$, fail to improve the model's generalized performance.

*A dataset is therefore* **contamination-resistant** *if its publicly released form remains useful at inference yet is unlearnable under current standard LLM training methods*. If data is

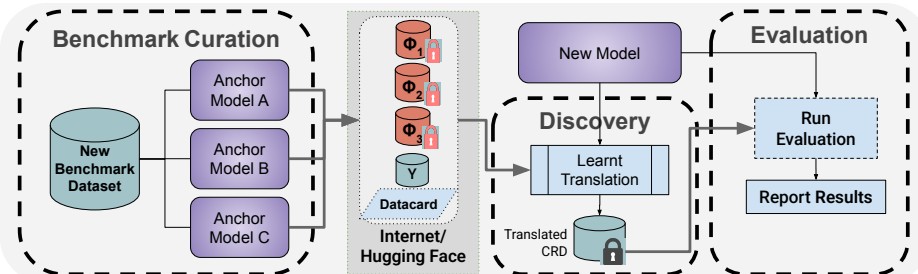

*Figure 2.* A contamination-resistant benchmark evaluation framework involves generating multiple projections at curation, and translation to the target model to be evaluated at discovery.

available in plaintext form, it is inherently vulnerable to being scraped and used for pretraining. As a result, for textual data to achieve this property, it has to be mapped into a latent form to be contamination-resistant.

In addition, we define three properties that CRDs must satisfy to achieve contamination resistance with respect to LLM benchmarking:

1. (Irreversibility) It should be computationally difficult or economically impractical at scale to reconstruct the original plaintext from the CRD, under realistic adversarial capabilities and contemporary training paradigms, i.e., given $\phi(\mathcal{D})$, it is difficult or non-economical to retrieve $\mathcal{D}$.

2. (Equivalence) The output of the model using the CRD should approximate the output of the original data, i.e., $\mathcal{M}(\phi(\mathcal{D})) \approx \mathcal{M}(\mathcal{D})$.

3. (Interoperability) Given $\phi(\mathcal{D})$, it should be possible to obtain $\phi_1(\mathcal{D})$ for another arbitrary LLM $\mathcal{M}_1$, such that $\phi_1(\mathcal{D})$ adheres to Property (1) and Property (2) (but may not itself adhere to this property).

These properties naturally follow from Definition 2.1. Property (1) ensures privacy by making recovery of the original plaintext computationally infeasible. Property (2) ensures practical utility: the dataset retains semantic structure for a pretrained model to perform inference and downstream tasks. Property (3) addresses a pragmatic requirement: it ensures the benchmark remains interoperable across various LLMs, as it would be infeasible to generate a separate representation for every individual model.

### 2.4. Evaluation Framework

Based on the definition and properties of CRDs, we provide an overview of a CRD LLM evaluation framework. Specifically, we address dataset curation, discovery, and model evaluation as depicted in Figure 2.

**Curation** Once the benchmark is prepared in plaintext, one or more representative *anchor models* are used to project the prompt or question portion of the test split into a latent representation. The choice of anchor models should be based on architectural alignment with modern LLM families, e.g., tokenization approach or attention mechanisms, to maximize the interoperability of the projection to target LLMs. We discuss interoperability in more detail in Section 4.

However, instead of releasing the entire latent projection, the release is limited to the specific form required for inference. This approach exploits the training–inference asymmetry in the Transformer architecture, and effectively prevents the benchmark from being used for training. The technical specifics are discussed in Section 3.

Each dataset is accompanied by a datacard that describes the provided latent projections and includes a representative subset of plaintext samples to facilitate translation for interoperability (see Section 4). The expected output ($Y$) is released in plaintext, since without access to the corresponding input the model cannot associate the solution with its question.

**Discovery** Before evaluation can take place, the CRD should be translated to the target model. Evaluators may opt for a *selective* translation of the most relevant projections or a *comprehensive* translation of all provided projections, depending on the computational budget and the required granularity of the assessment.

It is noteworthy that once a translation mapping is established between a specific target and anchor model, the transformation remains reusable. Any subsequent CRDs derived from the same anchor model can leverage the existing mapping, significantly reducing the overhead for future benchmarking tasks.

**Model Evaluation** To perform the evaluation, the translated representation is fed to the target model to generate continuations autoregressively. Throughout this process, the

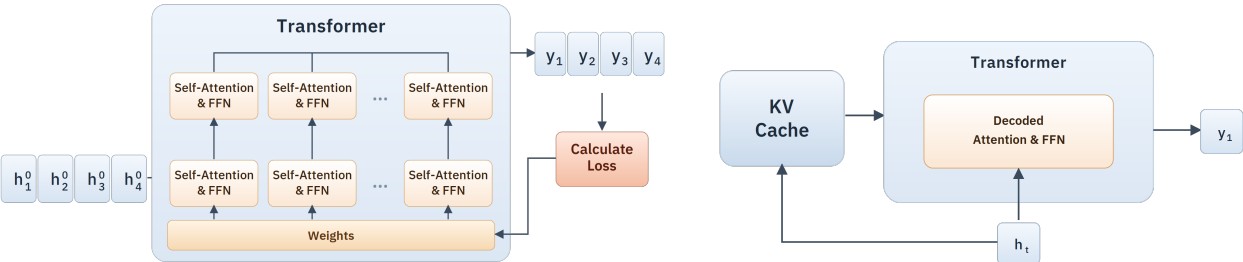

*(a)* Training: each token is used to calculate the loss for the position immediately after it

*(b)* Inference: KV Cache and penultimate layer hidden state are used to generate next token

*Figure 3.* CRDs rely on the asymmetry between the Training and Inference pipelines in the Transformer architecture

target model never has access to the plaintext benchmark questions.

The resulting outputs are then evaluated against the ground truth provided in the benchmark dataset. Since this ground truth is maintained in plaintext, it allows for seamless integration with standard scoring heuristics (e.g., Exact Match or semantic similarity).

## 3. Transformer Training/Inference Asymmetry

To support a CRD with respect to LLMs, we carefully examine the Transformer (Vaswani et al., 2017), which powers modern LLMs. In particular, we draw on the asymmetry in the Transformer's training and inference pipelines. Specifically, *training requires access to all tokens in a sequence to compute hidden states for each position and layer, while inference processes tokens sequentially and can leverage cached key-value pairs from previous tokens to avoid recomputation.* This motivates us to release benchmark datasets in a way that prevents them from being used for training, but remain usable for inference: we can release the KV cache and the penultimate (before-last) layer hidden state of the final token, without exposing the raw token sequences needed for training. This approach enables model evaluation on protected CRD benchmarks while preventing potential data contamination.

In other words, given a Transformer with $L$ layers, we demonstrate that releasing only the KV cache $\{K_{1:t}^{(l)}, V_{1:t}^{(l)}\}_{l=1}^{L}$ and the penultimate layer hidden state $h_t^{(L-1)}$ for each dataset input instance allows models to perform inference while preventing pretraining on the underlying token sequence.

**Transformer Training**   During training, the Transformer performs a next-token prediction task: given a sequence of tokens $x_1, x_2, \ldots, x_T$, the training objective is to minimize

the negative log-likelihood of the sequence.

$$\mathcal{L} = -\sum_{t=1}^{T} \log P(x_t \mid x_1, \ldots, x_{t-1}) \tag{1}$$

Each token $x_t$ is first mapped to an embedding vector, which is then passed through the stacked Transformer layers. In each layer $l$, the self-attention mechanism computes interactions between token $t$ and all tokens in positions 1 through $t$, generating a contextualized hidden state $h_t^{(l)}$. The feed-forward network (FFN) then applies a position-wise non-linear transformation to refine this representation before passing it to the next layer, as shown in Figure 3a. Finally, $h_t^{(L)}$ is projected into the vocabulary space to produce logits for predicting token $x_{t+1}$. Computing each hidden state requires access to the full prefix. Thus, *all tokens must be present during training* to compute the complete set of hidden states needed for learning contextual dependencies.

Similarly, fine-tuning requires processing the entire sequence to generate the hidden states needed for the label tokens, $Y$, even though loss is calculated only on $Y$.

**Inference Pipeline**   In contrast, during inference (see Figure 3b), the Transformer leverages KV caching using only two components: (1) the cached key-value pairs $K_{1:t}^{(l)}$ and $V_{1:t}^{(l)}$ for each layer $l = 1, \ldots, L$, and (2) the hidden state from the penultimate layer, $h_t^{(L-1)}$. All other intermediate hidden states from previous positions may be discarded.

To generate the first new token $y_1$, we begin with the cached state at position $t$. Given $h_t^{(L-1)}$, we compute the query for layer $L$:

$$Q_t^{(L)} = h_t^{(L-1)} W_Q^{(L)} \tag{2}$$

We then compute attention over the cached keys and values $K_{1:t}^{(L)}$ and $V_{1:t}^{(L)}$, and apply the remaining layer operations (output projection, residual connections, layer normalization, and feed-forward network) to obtain $h_t^{(L)}$. Applying

the language modeling head yields logits at position $t$, from which we sample $y_1 \sim \text{softmax}(\text{logits}_t)$.

To generate $y_2$, we embed $y_1$ as $h_{t+1}^{(0)} = \text{Embed}(y_1)$ and propagate through all $L$ layers: for each layer $l = 1, \ldots, L$, we compute queries, keys, and values from the current hidden state, append the new key-value pair to the cache, and compute attention over all cached positions 1 through $t + 1$. After applying residual connections, layer normalization, and feed-forward networks at each layer, we obtain $h_{t+1}^{(L)}$ and sample $y_2 \sim \text{softmax}(\text{logits}_{t+1})$. This demonstrates that inference requires only the KV cache and $h_t^{(L-1)}$. All other intermediate hidden states from previous positions can be discarded.

**Data Pair Format Definition**    Based on this, for a given data pair $(X, Y)$, we propose releasing the following:

$$\left( \{K_{1:t}^{(l)}, V_{1:t}^{(l)}\}_{l=1}^L, h_t^{(L-1)}, Y \right) \tag{3}$$

where $Y$ is released in plaintext and serves as the ground-truth label for evaluating model outputs.

## 4. Interoperability

In the previous section, we demonstrated that benchmark data may be released in a contamination-resistant form based on Transformer asymmetry. A natural objection arises: if benchmarks are encoded in the latent space of a specific model, how can heterogeneous models be evaluated against the same benchmark? Releasing separate encodings for every published LLM is clearly infeasible.

To address this limitation, we draw on recent advances in representation alignment. Specifically, we identify two potential paradigms: (1) a near-term *anchor-model* approach using subspace alignment techniques to translate between architectures; and (2) a longer-term *relative representation* approach that projects all models into a shared, model-agnostic coordinate frame. Both paradigms help advance contamination resistance (Section 2.3).

### 4.1. Background: Theoretical Foundations for Cross-Model Translation

Before presenting specific interoperability approaches, we establish the theoretical foundation that makes cross-model translation possible. Specifically, we consider three lines of research from the representation learning literature.

**The Platonic Representation Hypothesis.**    Huh et al. (2024) present empirical evidence that neural networks trained with different objectives on different data and modalities are converging toward a shared statistical model of reality in their representation spaces. As models scale and

become more general-purpose, the space of representations satisfying all task constraints shrinks, driving architecturally diverse models toward common solutions. The implication is significant: if representations are converging, then mathematical transformations between model-specific latent spaces should be increasingly accurate as models improve.

**Representational Similarity Metrics.**    Kornblith et al. (2019) introduce Centered Kernel Alignment (CKA) as a similarity index for comparing neural network representations. Their experiments demonstrate that architecturally identical networks trained independently develop layer-wise representations that CKA can accurately match. They provide quantitative evidence that *well-trained networks learn similar representations*. This suggests the geometric structure of latent spaces is more stable across training runs than previously assumed.

**Model Stitching.**    Bansal et al. (2021) revisit model stitching to study model internal representations. Given two trained models $A$ and $B$, they form a *stitched model* by connecting the bottom layers of $A$ to the top layers of $B$ through a simple trainable linear layer. Their experiments find that networks trained in very different ways (supervised vs. self-supervised, different architectures, different dataset sizes) can be stitched without significant performance degradation. This demonstrates that representations from distinct models can be functionally interchangeable under appropriate transformation.

Together, these findings establish that cross-model translation is grounded in empirically verified properties of neural representations. Based on this, we present two interoperability paradigms that build on these insights.

### 4.2. Near-Term Solution: Anchor Model with Subspace Alignment

This paradigm designates a widely adopted model (or models), $\mathcal{M}_{\text{anchor}}$, as the canonical reference for benchmark curation. Benchmark creators encode all evaluation items in the anchor model's latent format, specifically as key-value pairs ($KV_{\text{anchor}}$) and penultimate hidden state $h_{\text{anchor}}$. Model developers seeking to evaluate against the benchmark compute translation mappings from the anchor model to their target model's latent space.

Recent work on cross-model adapter transfer provides the underlying technical formulation for this approach. Xia et al. (2025) introduce Cross-LoRA, a framework for transferring LoRA modules between heterogeneous LLMs without access to training data. The method consists of two components: (a) *LoRA-Align*, which performs subspace alignment between source and target base models through rank-truncated singular value decomposition and Frobenius-

optimal linear transformation; and (b) *LoRA-Shift*, which projects the aligned source representations into the target model's parameter space.

LoRA-Align operates on the dominant singular subspaces of the anchor and target base weights, obtained via rank-$r$ truncated SVD. It then solves a pair of unconstrained least-squares problems to find linear maps $\hat{P}_U, \hat{P}_V$ that align the anchor's left and right singular bases with those of the target. The classical Procrustes problem (Schönemann, 1966) solves an analogous alignment but restricts the map to a rotation, which forces source and target to share the same dimension. Cross-LoRA relaxes this constraint to arbitrary linear maps, accommodating the dimension mismatch between heterogeneous models while retaining a closed-form solution. LoRA-Shift then projects an anchor-space update $\Delta W_{\text{anchor}}$ into the aligned subspace, producing a target-compatible update $\Delta W_t$ without any target adapter or training data. This same machinery motivates our extension to translating anchor-encoded benchmark representations ($KV_{\text{anchor}}$, $h_{\text{anchor}}$) into the target model's latent space.

Similarly, Wang et al. (2024) propose Trans-LoRA, which achieves higher-fidelity transfer through synthetic data generation and knowledge distillation. They demonstrate positive transfer (where the target model outperforms both the source adapter and its own base performance) across model families including LLaMA and Gemma. Farhadzadeh et al. (2025) introduce LoRA-X, which maintains compatibility within a shared column-row subspace, enabling training-free transfer for diffusion models and, more recently, language models.

Empirical studies reveal that translation fidelity (e.g., to preserve equivalence) depends on architectural similarity. Xia et al. (2025) report that models sharing grouped-query attention (GQA), SwiGLU activations, and RMSNorm normalization exhibit the strongest transferability, while differences in attention mechanisms or activation functions reduce but do not preclude successful alignment. This suggests that the anchor model should be chosen from a widely adopted architectural family to maximize equivalence.

Nonetheless, the subspace alignment approach preserves contamination resistance. Alignment matrices are computed solely from the weights of the two models, without access to the plaintext. Thereby, the CRD becomes interoperable while maintaining irreversibility.

### 4.3. Long-Term Vision: Model-Agnostic Relative Representations

The second paradigm eliminates model-specificity by projecting all latent spaces onto a shared coordinate frame. Rather than designating a canonical model, this approach defines representations relative to a fixed set of *anchor samples*

*ples*: inputs processed by all models to establish semantic alignment.

Moschella et al. (2023) introduce relative representations as a framework for zero-shot latent space communication. The key observation is that under consistent data and modeling choices, the angles between encodings within distinct latent spaces remain invariant even when the absolute coordinates differ arbitrarily. Formally, given an anchor set $\mathcal{A} = \{a_1, \ldots, a_k\}$, the relative representation of a point $x$ is the vector of similarities $[\text{sim}(x, a_1), \ldots, \text{sim}(x, a_k)]^T$. This projection maps any model's latent space into a common $k$-dimensional space where geometric relationships are preserved. The authors validate this approach across diverse settings including CNNs, vision transformers, graph neural networks, and text encoders, demonstrating zero-shot stitching (connecting an encoder from one model to a decoder from another) without additional training.

Maiorca et al. (2023) extend this framework to direct latent space translation via semantic alignment. Their method estimates an affine transformation between latent spaces using *parallel anchors*: sample pairs with established semantic correspondence. This enables closed-form transformations and zero-shot stitching across architectures (ResNet to ViT), domains, and modalities (text encoders to vision decoders) without retraining.

The relative representation approach offers several conceptual advantages over anchor-model alignment. First, it is inherently symmetric: all models are treated equally, and adding new models to the ecosystem requires only processing the shared anchor samples. Second, it naturally extends to cross-modal settings, potentially enabling benchmarks that span language, vision, and other modalities. Third, the theoretical grounding in angular invariance provides geometric interpretability that pure subspace projection lacks.

The anchor samples required for relative projection should not overlap with protected evaluation content. Instead, they can be drawn from public corpora, generated synthetically, or constructed from task-agnostic prompts. Moschella et al. (2023) demonstrate that effective alignment requires only a modest number of anchors (typically $k = 100$–$500$), and these anchors establish coordinate correspondence without encoding benchmark-specific information. Consequently, relative projection supports interoperability while preserving irreversibility and providing more consistent equivalence.

## 5. Discussion

In this section, we evaluate how well our definition of contamination resistance is realized by the methods presented in Section 3 and Section 4, and how existing advances in the literature can mitigate or address shortcomings.

**(Property 1) Irreversibility**    Prior work (Wu et al., 2025a; Luo et al., 2026) has shown that inversion attacks can recover input information directly from KV cache representations. Importantly, these attacks remain effective even when the KV cache is compressed or quantized, which are common optimizations used to reduce memory usage and improve inference speed. However, the effectiveness of these attacks depends heavily on the underlying attention architecture. Luo et al. (2026) find that inversion is feasible for models using Multi-Head Attention (MHA), where attention weight matrices can be inverted, but is much less effective on models using Grouped-Query Attention (GQA) and related modern architectures. Since many current LLMs no longer rely on standard MHA, these attacks do not generalize broadly across deployed systems.

Furthermore, model representations are lossy abstractions that map multiple possible plaintext representations into similar embedding spaces, which makes the recovery of the original plaintext computationally difficult even when inversion is theoretically possible.

To preserve irreversibility, specialized defensive mechanisms may be required to mitigate potential inversion attacks. Common defenses against leakage involve adding calibrated noise to the output distributions (Mattern et al., 2023), applying entropy-based perturbations (Jin et al., 2025), or incorporating differential privacy mechanisms (Yu et al., 2022). On the other hand, compression approaches have been proposed to compress the KV cache into more secure formats (Ge et al., 2024; Chu et al., 2025), or utilizing methods such as KV-Cloak (Luo et al., 2026) that obfuscate KV matrices while maintaining their utility.

Nonetheless, the definition of irreversibility depends on the application and threat model. While in many practical settings it is enough that reconstructing the original input is computationally infeasible, in more privacy-sensitive settings (e.g., releasing benchmarks derived from user data) stronger guarantees may be needed to limit information leakage. In these cases, additional measures such as withholding anchor model weights may also be appropriate. As a result, a third party that owns the anchor model may provide an encoding service via open or paid APIs, which is a far less involved process than complete private benchmarking, and is only required once per model-dataset combination.

**(Property 2) Equivalence**    Unlike open benchmarks, where researchers can read and verify the evaluation examples, this approach introduces opacity that creates trust concerns: how can the community verify that the released KV representations preserve the evaluation properties of the original benchmark?

To ensure equivalence, benchmark releases can include reproducible verification protocols, such as paired evaluations against an anchor model, calibration checks across model families, or distributional tests over outputs (Siska et al., 2024), that allow the community to audit whether the transformed benchmark faithfully preserves evaluation utility. Additionally, to ensure that benchmark scores more accurately reflect a model's underlying capabilities, it may be useful to backtest traditional evaluation methods against their CRD counterparts by comparing performance on the original and encoded versions of existing benchmarks.

**(Property 3) Interoperability**    Intuitively, translating KV caches from an anchor model to a target model is likely to introduce accumulated errors over long generation sequences, potentially causing the translated caches to drift from those produced under native inference and thereby degrading generation quality. However, prior work on cross-model representation transfer, such as model stitching (Bansal et al., 2021), has shown only limited degradation, as discussed in Section 4. More recently, researchers have empirically observed overall performance improvements using cache-to-cache translation mechanisms (Fu et al., 2026). In both cases, these approaches enable accelerated inference while substantially reducing the computational cost of large-scale evaluation.

To better reflect their plaintext performance, several methodologies may be used for fine-grained calibration of model outputs. These can include: (1) *distillation techniques* to ensure the translated prefix closely aligns with the target model's native activation distribution (Hinton et al., 2015), (2) *prompt tuning* to normalize results against the model's output distribution using a prefix to filter out systematic biases introduced by translation (Li & Liang, 2021), (3) *adaptive temperature scaling* to estimate variance and stabilize autoregressive generation (Xie et al., 2024).

**Benchmark Compatibility**    CRDs are broadly compatible with benchmarks where inputs and outputs remain structurally independent. Compatible benchmarks include: (1) single-turn question-answering tasks (e.g., MMLU (Hendrycks et al., 2021b), SQuAD (Rajpurkar et al., 2016), HumanEval (Chen et al., 2021)), where each instance consists of discrete question-answer pairs; (2) classification and labeling benchmarks (e.g., GLUE (Wang et al., 2018), SuperGLUE (Wang et al., 2019), ImageNet (Russakovsky et al., 2015)), where test instances have clear input-output boundaries; (3) multi-modal benchmarks (e.g., COCO (Lin et al., 2014), Flickr30K (Young et al., 2014)) where images or other modalities serve as inputs and captions or annotations as outputs; (4) code generation benchmarks (e.g., CodeContests (Li et al., 2022), APPS (Hendrycks et al., 2021a)), where problem statements are inputs and solutions are outputs; and (5) summarization benchmarks (e.g., CNN/DailyMail (See et al., 2017), XSum (Narayan et al.,

2018)), where articles are inputs and summaries are outputs.

On the other hand, benchmarks become less compatible when inputs and outputs are tightly coupled or when context evolves across turns. Incompatible or partially incompatible benchmarks include: (1) multi-turn conversational benchmarks (e.g., CoQA (Reddy et al., 2019), QuAC (Choi et al., 2018), MultiWOZ (Budzianowski et al., 2018)), where later turns depend on or modify earlier context, entangling inputs and outputs; (2) highly dynamic agentic benchmarks (e.g., WebShop (Yao et al., 2022), ALFWorld (Shridhar et al., 2021)), where agent actions feed back into the environment and reshape subsequent observations; (3) interactive or adaptive benchmarks (e.g., DynaBench (Kiela et al., 2021), AdaTest (Ribeiro & Lundberg, 2022)), where test instances are modified based on model outputs; and (4) benchmarks with implicit or evolving gold standards (e.g., GAIA (Mialon et al., 2024), MLE-bench (Chan et al., 2025)), where correctness depends on the trajectory rather than isolated input-output pairs. For some of these benchmarks, CRD methodology would require careful adaptation or may be better suited to specific sub-components of the evaluation rather than the full benchmark.

**Space Complexity**   A key practical consideration is the storage overhead of releasing KV caches. For example, a KV cache for 100K tokens in LLaMA-2 7B requires 50GB of disk space (Cai et al., 2025). However, this overhead remains manageable: quantization and sparse encoding can reduce KV cache footprints significantly with negligible impact on evaluation fidelity (Jin et al., 2024). Zhang et al. (2023) show that retaining only 20% of the KV cache preserves comparable performance, while PyramidKV (Cai et al., 2025) has demonstrated that 12% suffices to maintain performance, and even 0.7% has only a subtle effect. Additionally, we can selectively drop KV entries for tokens that provide limited task-relevant information, such as formatting tokens, generic instructions, or boilerplate text, while retaining entries for semantically critical content to further reduce storage requirements. This effectively reduces the storage requirement to 350 MB for 100K tokens. Figure 4 presents disk space requirements for various benchmarks using PyramidKV.

**Limitations**   While we provide a general mechanism, its practical applicability may depend on architecture-specific factors. CRDs hinge on LLMs implementing the Transformer architecture, and as a result they do not apply to non-Transformer-based models, such as Mamba (Gu & Dao, 2024). Furthermore, variations in Transformer-based model families (e.g., GPT, LLaMA), attention mechanisms, positional encoding strategies, and related design choices can introduce nontrivial differences in behavior, performance, and security properties. Addressing these interactions and

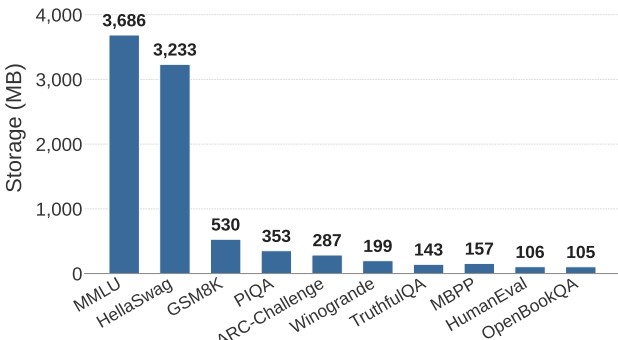

*Figure 4.* Approximate storage requirements for CRD projection of various existing benchmarks calculated based on number of test questions and average question token size using Llama2-7B and PyramidKV compression

identifying optimal adaptations for each architecture is beyond the scope of this work.

## 6. Alternative Views

Alternative approaches to benchmark contamination fall into two categories (Xu et al., 2024): *Data Curation*, which uses data unavailable during training (either private or published after pretraining), and *Data Refactoring*, which modifies existing benchmarks through rephrasing or updates to mitigate memorization, or filters out samples known to be in training corpora. We examine both approaches below.

### 6.1. Data Curation

Data curation methods attempt to safeguard benchmarking integrity by either shielding benchmark datasets from the model's view or modifying them to ensure that the LLM has not been previously trained on specific instances. The two primary strategies for achieving this are *private benchmarking*, which utilizes *private* datasets to evaluate models behind closed doors (Rajore et al., 2024), and *dynamic benchmarking*, which periodically refreshes or modifies existing datasets by introducing novel data that could not have been exposed to the model, thereby creating a moving target for evaluation (Wu et al., 2025b; Zhao et al., 2025).

**Private Benchmarking**   Private benchmarking uses a trusted third party to evaluate models on private benchmarks. Developers submit their weights or provide API access to a centralized authority for evaluation (Rajore et al., 2024). TRUCE (Rajore et al., 2024) facilitates this using multiple configurations: either a trusted dataset owner hosts the LLM and is trusted to benchmark it correctly, or a secure computation environment is provided to combine models and architectures for on-demand benchmarking. TRUCE provides secure and verifiable communication channels be-

tween participants and includes dataset auditing to ensure dataset quality.

While private benchmarking creates a *black-box* environment that protects data, it introduces significant bottlenecks. It can limit testing frequency and create a cost barrier that disadvantages open-source contributors. Furthermore, sharing weights is often infeasible for proprietary models, while API-based evaluation risks exposing the dataset.

**Dynamic Benchmarking** Dynamic benchmarking transforms evaluation into a moving target by periodically generating new, unseen test instances. This approach often utilizes information or knowledge that postdates the model's training. For example, AntiLeakBench (Wu et al., 2025b) automatically constructs evaluation sets by pulling updated real-world knowledge from the internet. This ensures the test data is chronologically *newer* than the model's training cutoff. Likewise, CoreEval (Zhao et al., 2025) updates datasets by replacing outdated information with current real-world knowledge while maintaining semantic coherence.

However, the utility of these new datasets is often short-lived, as they are rapidly absorbed into pretraining corpora after public release. This cycle of immediate ingestion precludes the establishment of stable, high-quality benchmarks required for meaningful longitudinal evaluation and tracking model progress over time.

### 6.2. Data Refactoring

Data refactoring transforms benchmark content to reduce its utility as training data while maintaining its utility as an evaluation signal, by disrupting the mapping between input (question) and output.

**Lexical Obfuscation** Lexical obfuscation involves rephrasing or perturbing dataset wording to prevent models from relying on memorization. Common techniques involve shuffling multiple-choice options (Pezeshkpour & Hruschka, 2024), substituting synonyms (Emmery et al., 2021), or applying reversible ciphers to rename code elements (Ni et al., 2023).

However, recent research has shown that modern LLMs often circumvent these transformations because they have already been trained on the obfuscated formats themselves (Samuel et al., 2025). The newly created obfuscation results in additional contamination that is even more resistant to lexical changes.

**Decontamination and Filtering** Decontamination and filtering techniques aim to remove benchmark samples from pretraining corpora. Common approaches include n-gram overlap detection, hash-based matching, exact string filtering, and embedding-based semantic similarity searches to identify contaminated data (Dodge et al., 2021; Magar & Schwartz, 2022).

While these methods can substantially reduce direct leakage, their effectiveness is fundamentally limited. Exact or near-exact matching fails to capture paraphrased, translated, or lightly transformed content. Semantic filtering introduces trade-offs between recall and precision that may result in excessive data removal. Moreover, decontamination pipelines are typically applied at a fixed point in time and do not account for indirect leakage through continual pretraining, model distillation, or synthetic data generation from contaminated models (Yang et al., 2023). As training corpora grow in scale and heterogeneity, these limitations make reliable decontamination increasingly difficult and costly.

## 7. Conclusion

We advocate for benchmarking methodologies that explicitly account for the inevitability of dataset contamination in large-scale language model training. Rather than assuming perfectly held-out evaluation data, benchmarks should be released in a contamination-resistant format that maintains inference utility but is unlearnable, while providing irreversibility, equivalence, and interoperability.

We leveraged the training–inference asymmetry in Transformers and proposed two interoperability frameworks: an anchor-model-based approach that can facilitate near-term adoption and a longer-term approach that can provide more consistent benchmarking. Furthermore, while CRDs increase disk space requirements, the overhead remains reasonable.

We call on the community to: (i) advance the CRD ecosystem by developing scalable architectures and evaluation methodologies, including testing existing methods, such as model stitching, for low-friction adoption; (ii) establish a standardized set of anchor models as universal encoders, reducing the computational burden of projection layers for new LLM releases; (iii) integrate CRD-based validation into existing pipelines (e.g., Hugging Face), streamlining benchmarking requirements for individual users while making CRDs a seamless component of experimentation and deployment.

Together, these directions outline a practical path toward fair LLM benchmarking. By lowering integration overhead while improving comparability, CRDs have the potential to make model evaluation more consistent and reliable without jeopardizing efficiency and usability.

## Acknowledgments

This material is based upon work supported by, or in part by, the Army Research Office (ARO) under grant num-

ber W911NF-21-1-0198 and in part by U.S. NSF awards #2114824 and #2438810.

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
