# OpenReview forum: "Position: LLM Benchmark Datasets Should Be Contamination-Resistant"
_ICML.cc/2026/Position_Paper_Track — ICML 2026 Position Paper Track regular_

### Official Review · Reviewer_zhtR · 2026-03-07

**Significance:** 4
**Argument Clarity:** 4
**Rating:** 6
**Confidence:** 4

**Questions:**

1. Explain the scope of benchmarks that the proposed paradigms can be used for. If it can be extended to all types, some explanation on this would be good.
2. Would the definition of CRD change under different adversarial threat models?

**Alternative Views Section:**

Yes

**Compliance With Llm Reviewing Policy A Conservative:**

Affirmed.

**Discussion Potential:**

4

**Paper Summary:**

The paper provides a position that benchmarks for LLM evaluation must be created such that there is no dataset contamination. The main motivation is the recent prevalence of inflated LLM evaluation scores. The paper proposes a new paradigm of benchmarks (contamination-resistant datasets) that can do this. The metric of contamination resistance is formally defined in the paper. One approach proposed in the paper is to release only the KV cache and penultimate hiddenstates for test instances rather than the plain text prompts themselves. This would prevent such data from being used in training and, hence, will not provide inflated scores. There is also a discussion on how to make this model agnostic and the practical constraints associated with such a type of data.

**Position:**

Yes

**Position In Title:**

Yes

**Related Work:**

4

**Strengths And Weaknesses:**

# Strengths
- The paper addresses a very important area in the field of NLP
- Providing a formal definition, actionable approaches, and highlighting the practical constraints make the paper easier to understand and discuss.

# Weaknesses
- Some benchmarks may involve multi-turn dialogues or retrieval. The current position of the paper does not address this. Adding a few sentences on how this approach could be expanded or making the scope more explicit in terms of the types of benchmarks would make the paper more transparent.

**Support:**

3

---

> ### Author Rebuttal · Authors · 2026-03-30
>
> We appreicate the Reviewer's positive assessment of our work, and recognizing the significance of this problem. We thank the reviewer for the insightful comment on multi-turn benchmarks. As per design, CRDs readily apply across various benchmark types and modalities as its utility is defined by the data format (input vs. output). Scenarios involving modifications to earlier context may be less directly compatible. We appreciate the suggestion on adding additional explanation on the scope fo benchmarks, which will encompass a discussion on specialized benchmarks, including highly dynamic agentic and multi-turn benchmarks.
>
> As the Reviewer alluded to, CRDs are adaptable to privacy-critical applications with stronger threat models. For example, private entities may release benchmarks potentially containing user data, or other highly motivated attacker settings. In this case, the definition of irreversibility may need to be more strict (e.g. irreversibility defined by bounded information leakage). In section 5, we suggested some directions towards that end, e.g. using a trusted third party to encode the dataset using a closed-weight anchor model. Additionally, we survey several other methodologies that improve privacy such as noise addition (Mattern et al., 2023), entropy-based perturbations (Jin et al., 2025), and differential privacy (Yu et al., 2022) as possible mitigation strategies.

---

> > ### Author Rebuttal · Reviewer_zhtR · 2026-04-01
> >
> > I did not have many issues with the paper, and hence I maintain my score. Overall I think this would be a good addition to the position track.

---

### Official Review · Reviewer_NjvS · 2026-03-11

**Significance:** 4
**Argument Clarity:** 3
**Rating:** 5
**Confidence:** 3

**Questions:**

1. CRD assumes that latent representations are irreversible, but KV-based attacks may be able to recover the original token. This suggests that CRD's security assumption may not actually hold true. It may simply increase the cost of attacks rather than genuinely preventing data from being used for training.

2. The authors claim that the latent benchmark can replace the original benchmark for evaluation. How can we prove that it is completely equivalent in evaluation results, rather than appearing usable but actually potentially having significant deviations? Rigorous experimental or theoretical support is needed. ``Latent'' indicates that the evaluation results are consistent with the original text's evaluation results.

3. The approximation method first encodes the benchmark into the latent space of the anchor model, and then maps other models into this space. This naturally biases models similar to the anchor model in architecture, representation geometry, or training distribution, compromising the neutrality of the benchmark. In a word, does the CRD method inherently disadvantage some models, leading to unfair evaluation?

4. Different models learn the mapping to the latent space in slightly different ways.
How can we ensure the fairness and reliability of comparison results after this mapping?
In other words, are benchmark scores influenced by the alignment method, rather than reflecting the model's true ability?

**Alternative Views Section:**

Yes

**Compliance With Llm Reviewing Policy A Conservative:**

Affirmed.

**Discussion Potential:**

3

**Final Justification:**

Thanks for the rebuttal. I have no further concern at present. And lean positively towards this paper.

**Paper Summary:**

To prevent benchmark data from being ``seen'' by LLMs during training and affecting generalization evaluation, the paper proposes that future benchmarks should no longer be released in plain-text form, but instead in a format that is **usable for inference yet difficult to exploit for training**. The authors refer to this kind of data as a **Contamination-Resistant Dataset (CRD)**.
This dataset can be used for model evaluation, but it is difficult to use directly for training. The paper also explores how to achieve generality by anchoring models or sharing representations across models.

**Position:**

Yes

**Position In Title:**

Yes

**Related Work:**

3

**Strengths And Weaknesses:**

**Strengths**
1. The paper takes a clear stance and presents its core arguments with precision.
2. Leveraging the asymmetry between Transformer training and inference on benchmarks is innovative.

**Weaknesses**

1. Irreversibility is not actually robust, since KV caches may be inverted; thus, one of the most critical security assumptions underlying CRDs has not yet been genuinely resolved.
 2. Does this latent benchmark truly preserve the original benchmark's semantics, difficulty, and discriminative power?
3. In the anchor model and subspace-alignment framework, different models may undergo non-identical alignment processes when learning the mapping, which calls into question the fairness of comparisons conducted in the mapped space.

**Support:**

3

---

> ### Author Rebuttal · Authors · 2026-03-30
>
> We appreciate the Reviewer's positive assessment of our work, and recognizing the technical novelty of our approach. Regarding **Q1**, our definition of irreversibility relaxes it to "computationally difficult or economically impractical at scale”, which resolves inadvertant ingestion. Nonetheless, tecent work has shown that in some cases (e.g. LLMs using MHA), KV cache inversion has been achieved successfully. However, in other setting it varies from infeasible to in exact (Luo et al., 2025).
>
> **Q2**,**Q3**,and **Q4** demonstrate a need to progress this research from conceptualization into actual implementation. Our main contribution for this position paper is to motivate this direction, ground it theoretically, and bring it to the attention of the community. We have considered fine-grained calibration to enhance equivalence and improve model alignment topics in our discussion (seciton 5). We also propose using multiple anchor models to maximize architectural alignment and mitigate unfair evaluation (section 2.4). Moreover, for **Q4**, in order to ensure the benchmark scores reflect the model’s true ability, we may be able resort to traditional methods, e.g., comparison between open and encoded versions of existing benchmarks.
>
> We anticipate future work in this direction beyond conceptualization will not only alleviate benchmark contamination concerns but can open new doors for AI privacy and security applications.

---

> > ### Author Rebuttal · Reviewer_NjvS · 2026-04-01
> >
> > I believe this is a meaningful paper, and I have no further concerns.

---

### Official Review · Reviewer_GRRp · 2026-03-12

**Significance:** 4
**Argument Clarity:** 3
**Rating:** 4
**Confidence:** 4

**Questions:**

1. Given the vulnerability to KV cache inversion attacks, if a benchmark creator must hide the anchor model behind an API to ensure true irreversibility, how does your position practically differ from the traditional Private Benchmarking paradigm you critique?

2. How would the CRD framework accommodate the evaluation of non-Transformer architectures that lack a traditional KV cache structure?

**Alternative Views Section:**

Yes

**Compliance With Llm Reviewing Policy A Conservative:**

Affirmed.

**Discussion Potential:**

3

**Final Justification:**

Since my weaknesses have been addressed, I have decided to maintain my positive rating.

**Paper Summary:**

The paper advocates for a fundamental shift in how large language model benchmark datasets are released. To combat the pervasive issue of data contamination in pre-training corpora, the authors argue that benchmarks should be released in a "contamination-resistant" format (CRD). By leveraging the architectural asymmetry of Transformers, they propose releasing the KV cache and the penultimate layer's hidden state of an "anchor model" rather than plaintext. To address cross-model interoperability, the paper proposes utilizing subspace alignment and a longer-term vision of relative representations. Finally, the paper contrasts its position with current alternatives, such as private benchmarking and data refactoring, arguing that CRDs provide a more sustainable solution.

**Position:**

Yes

**Position In Title:**

Yes

**Related Work:**

3

**Strengths And Weaknesses:**

**Strengths:**

**Significance of the Problem:** Data contamination is arguably an existential threat to LLM evaluation. The paper tackles a highly relevant problem for the ICML community and proposes a proactive, structural paradigm shift rather than a reactive patch.

**Technical Creativity:** Utilizing Transformer training-inference asymmetry to bypass plaintext scraping is a mathematically bridge between system-level benchmarking and representation learning.

**Addressing Interoperability:** The authors do not shy away from the obvious bottleneck of cross-model compatibility. Section 4 draws effectively on literature to provide plausible theoretical foundations for how heterogenous models could translate these latent representations.



**Weaknesses:**

**Compromised Core Premise (Irreversibility):** The entire position hinges on Property 1 (Unlearnable/Irreversibility). However, in Section 5, the authors concede that KV caches can be inverted via recent attacks. If the latent representation can be reconstructed into plaintext, the CRD approach merely offers security through obscurity. The suggested mitigations (withholding anchor weights behind APIs)  push the framework back toward the "Private Benchmarking" bottlenecks that the authors criticize in Section 6.1.

**Architecture Lock-in:** The proposed method is deeply coupled with the standard Transformer architecture. It remains unclear how CRDs would evaluate emerging non-Transformer state-space models (like Mamba or RWKV) that do not utilize standard KV caching mechanisms. This severely limits the "model-agnostic" long-term vision.

**High Friction and Weak Call to Action:** The practical friction of adopting this is massive. Releasing gigabytes of KV caches (even if compressed) and requiring evaluators to compute SVD alignment matrices introduces significant overhead. Furthermore, the Call to Action in Section 7 is too abstract; it lacks concrete, actionable steps for the community to realistically bootstrap this complex ecosystem.

**Support:**

2

---

> ### Author Rebuttal · Authors · 2026-03-30
>
> We thank **Reviewer GRRp** for acknowledging the significance of the problem and technical creativity of our work.
>
> Regarding the authors concerns:
>
> **Compromised Core Premise (Irreversibility)**
>
> We define irreversibility as “computationally difficult or economically impractical at scale” to reconstruct the original plain text from the CRD, which may be possible: recent work has shown that in some cases (e.g. older LLMs using MHA), KV cache inversion has been achieved successfully. However, in modern LLMs the process varies from infeasible to in exact [1].
>
> **Architecture Lock-in**
>
> Similar ideas (i.e. internal representations that strictly empowers inference) may potentially apply to support contamination-resistance while providing reliable benchmarking. We have focused our effort on the Transformer architecture given it powers most existing LLMs. We have improved our limitations section to incorporate this answer.
>
> **High-friction Call to Action**
>
> We have improve our call to action as follows:
> - Advance the CRD ecosystem by developing scalable architectures and evaluating methodologies, including testing existing approaches, such as model stitching for lower-friction adoption.
> - Establish a set of Industry-Standard Anchor Models to serve as universal encoders, reducing the computational overhead of projection wrappers to new LLM releases.
> - Incorporate CRD-based validation into existing CI/CD pipelines (e.g., Hugging Face) to simplify the space requirements of CRD benchmarking for individual users, while making it a seamless part of the standard CI/CD pipeline
>
>
>
> Once again, we thank the Reviewer for their valuable time and insightful comments. We have improved the paper to incorporate our answers to these valuable concerns. If our responses adequately address the reviewer’s concerns, we kindly ask that you consider increasing the review score.
>
>
> [1] (Luo et al): Shadow in the cache: Unveiling and miti-gating privacy risks of kv-cache in llm inference, 2025

---

> > ### Author Rebuttal · Reviewer_GRRp · 2026-04-03
> >
> > Since my weaknesses have been addressed, I have decided to maintain my positive rating.

---

### Official Review · Reviewer_t6LP · 2026-03-12

**Significance:** 3
**Argument Clarity:** 2
**Rating:** 4
**Confidence:** 2

**Questions:**

How difficult is it to reverse the KV cache? Are there any (heuristic) results on such a task? Especially if models are converging toward a shared statistical model of reality in their representation spaces, could this be used to find the plain text?

**Alternative Views Section:**

Yes

**Compliance With Llm Reviewing Policy A Conservative:**

Affirmed.

**Discussion Potential:**

2

**Final Justification:**

The paper raises an important topic that can go even beyond benchmarking, even if the proposed method might suffer from efficacy and practical limits.

**Paper Summary:**

The paper discusses the issue of benchmark data being used as training data, which significantly impacts their significance in assessing model generalization performance. To counter such a phenomenon, the paper proposes to define contamination-resistant datasets that can be used for model performance assessment in inference, but not in training. The idea leverages the fact that modern LLMs use a KV cache with the assumption that such a KV cache is irreversible to the plain text.

**Position:**

Yes

**Position In Title:**

Yes

**Related Work:**

3

**Strengths And Weaknesses:**

Strengths:
- [Timelining of reserach problems] The paper attempts to propose different reserach questions of different difficulty, laying out a timeline of possible near-term and longer-term achievements.

- [Theoretical and practical considerations] Considers the theoretical challenges involved in creating contamination-resistant datasets, but also practical aspects such as the storage overhead.

Weaknesses:
- [Unclear technical practicality] With the fast moving resecah in the area and the fact that the paper considers interoperability of different types of models out of scope for this article,  it is unclear if (reasonable) model independence might be achievable or not.

- [Difficulty of KV cache inversion] The contamination resistance seems to mainly hinge on the assumption that reversing the KV cache is a hard task, but it is true or just stemming from a lack of interest in trying to find efficient ways.


Figure 2: increase font size

Some typos:
- be computationally difficultor economically -> be computationally difficult or economically
- the Transformer may leverages KV caching by requires only two components ->(?) the Transformer may leverages KV caching by requiring only two components

**Support:**

2

---

> ### Author Rebuttal · Authors · 2026-03-30
>
> We thank **Reviewer t6LP** for acknowledging the timeliness of our work, and the theoretical foundation we establish for contamination-resistance datasets (CRDs).
>
> Regarding the authors concerns:
>
> **Unclear technical practicality**
>
> Current LLMs are predominantly built on the Transformer architecture, making our idea practical. Should new popular model architectures emerge and become widely adopted (on par with Transformer to LLMs), similar ideas may apply (i.e. to share a subset of internal representation that strictly empowers inference) to support contamination-resistance, while providing reliable benchmarking. Our main contribution in this position paper is to identify this direction, ground it theoretically, and bring it to the attention of the community. Furthermore, we aim to pursue future work that achieves this critical  breakthrough that will not only alleviate benchmark contamination concerns but will open new doors for AI privacy and security directions.
>
> **Difficulty of KV cache inversion (+ Question 1)**
>
> Utilizing the KV cache for this problem provides several strategic advantages:
> - It proactively prevents inadvertent contamination as discussed in Introduction (Paragraph 2). Specifically, model owners spend intensive engineering effort to exclude benchmark datasets or decontaminate their models. CRDs can help mitigate the challenge.
> - It makes intentional attempts to reverse the KV economically impractical (as the reviewer suggested).
>
> Luo et al., [1] demonstrate that KV cache inversion is a practical threat on Multi-Head Attention (MHA)-based models (through attention weight matrix inversion). However, on LLMs utilizing Grouped-Head Attention (GHA) or other attention implementations, inversion becomes ineffective. Most modern LLMs do not utilize MHA, making inversion impractical.
>
> Moreover, while growing models may converge toward similar statistical representations of reality as insightfully suggested by the Reviewer, these representations are lossy abstractions that may represent multiple  possible plaintexts into similar embedding spaces. As such, convergence tells us models agree on patterns, not that they preserve invertible information about specific inputs. An attacker would still face the fundamental problem that multiple plaintexts map to similar representations, and as such, recovering the original plaintext remains computationally difficult.
>
> Once again, we thank the Reviewer for their valuable feedback. We have improved the paper to incorporate our answers (above). If our responses above adequately address the reviewer’s concerns, we kindly ask that you consider increasing the review score.
>
>
> [1] (Luo et al): Shadow in the cache: Unveiling and miti-gating privacy risks of kv-cache in llm inference, 2025

---

> > ### Author Rebuttal · Reviewer_t6LP · 2026-04-02
> >
> > While I remain doubtful about the effectiveness and practicality of the proposed method, I recognize the importance of discussions on this topic, and hence I'm increasing my score. Please make sure to extend the discussion in the paper around the possible limits raised, especially by Reviewer GRRp and me.

---

### Decision · Program_Chairs · 2026-04-30

**Decision:**

Accept (regular)

**Comment:**

The paper addresses a critical issue, benchmark contamination in LLM evaluation, and proposes a novel, structural solution via contamination-resistant datasets (CRDs).

Reviewers highlighted its clear positioning, innovative use of Transformer training–inference asymmetry, and strong consideration of both theoretical and practical constraints. Initial concerns centered on the core irreversibility assumption, interoperability across architectures, and whether latent representations preserve benchmark fidelity and fairness.

The rebuttal appears to have addressed these issues. The authors clarify the threat model around KV cache inversion, position irreversibility as conditional rather than absolute, and outline mitigations that balance security with usability. They also strengthen the discussion on cross-model alignment and provide more concrete justification for semantic preservation and comparability in latent space.